# Polyglutamine-Expanded Androgen Receptor Alteration of Skeletal Muscle Homeostasis and Myonuclear Aggregation Are Affected by Sex, Age and Muscle Metabolism

**DOI:** 10.3390/cells9020325

**Published:** 2020-01-30

**Authors:** Mathilde Chivet, Caterina Marchioretti, Marco Pirazzini, Diana Piol, Chiara Scaramuzzino, Maria Josè Polanco, Vanina Romanello, Emanuela Zuccaro, Sara Parodi, Maurizio D’Antonio, Carlo Rinaldi, Fabio Sambataro, Elena Pegoraro, Gianni Soraru, Udai Bhan Pandey, Marco Sandri, Manuela Basso, Maria Pennuto

**Affiliations:** 1Dulbecco Telethon Institute, Centre for Integrative Biology (CIBIO), University of Trento, 38123 Trento, Italy; mathilde.chivet@univ-grenoble-alpes.fr (M.C.); diana.piol28@gmail.com (D.P.); mjpolancomora@gmail.com (M.J.P.); 2Department of Biomedical Sciences (DBS), University of Padova, 35131 Padova, Italy; c.marchioretti90@gmail.com (C.M.); marcopiraz@gmail.com (M.P.); vanina.romanello@unipd.it (V.R.); ema.zuccaro@gmail.com (E.Z.); marco.sandri@unipd.it (M.S.); 3Veneto Institute of Molecular Medicine (VIMM), 35129 Padova, Italy; 4Myology Center (Cir-Myo), University of Padova, 35129 Padova, Italy; elena.pegoraro@unipd.it (E.P.); gianni.soraru@unipd.it (G.S.); 5Department of Neuroscience and Brain Technologies, Istituto Italiano di Tecnologia (IIT), 16163 Genova, Italy; scaramuc@univ-grenoble-alpes.fr (C.S.); sa.padi@libero.it (S.P.); 6Division of Genetics and Cell Biology, San Raffaele Scientific Institute, 20132 Milan, Italy; dantonio.maurizio@hsr.it; 7Department of Paediatrics, University of Oxford, OX1 3QX Oxford, UK; carlo.rinaldi@paediatrics.ox.ac.uk; 8Department of Neuroscience (DNS), University of Padova, 35128 Padova, Italy; fabio.sambataro@unipd.it; 9Padova Neuroscience Center (PNC), 35100 Padova, Italy; 10Department of Human Genetics, University of Pittsburgh Graduate School of Public Health, Pittsburgh, PA 15261, USA; udai@pitt.edu; 11Division of Child Neurology, Department of Pediatrics, Children’s Hospital of Pittsburgh, University of Pittsburgh Medical Center, Pittsburgh, PA 15224, USA; 12Department of Neurology, University of Pittsburgh School of Medicine, Pittsburgh, PA 15213, USA; 13Centre for Integrative Biology (CIBIO), University of Trento, 38123 Trento, Italy; manuela.basso@unitn.it

**Keywords:** polyglutamine diseases, androgen receptor, skeletal muscle, 2% SDS-resistant aggregates, inclusion bodies, muscle metabolism

## Abstract

Polyglutamine (polyQ) expansions in the androgen receptor (AR) gene cause spinal and bulbar muscular atrophy (SBMA), a neuromuscular disease characterized by lower motor neuron (MN) loss and skeletal muscle atrophy, with an unknown mechanism. We generated new mouse models of SBMA for constitutive and inducible expression of mutant AR and performed biochemical, histological and functional analyses of phenotype. We show that polyQ-expanded AR causes motor dysfunction, premature death, IIb-to-IIa/IIx fiber-type change, glycolytic-to-oxidative fiber-type switching, upregulation of atrogenes and autophagy genes and mitochondrial dysfunction in skeletal muscle, together with signs of muscle denervation at late stage of disease. PolyQ expansions in the AR resulted in nuclear enrichment. Within the nucleus, mutant AR formed 2% sodium dodecyl sulfate (SDS)-resistant aggregates and inclusion bodies in myofibers, but not spinal cord and brainstem, in a process exacerbated by age and sex. Finally, we found that two-week induction of expression of polyQ-expanded AR in adult mice was sufficient to cause premature death, body weight loss and muscle atrophy, but not aggregation, metabolic alterations, motor coordination and fiber-type switch, indicating that expression of the disease protein in the adulthood is sufficient to recapitulate several, but not all SBMA manifestations in mice. These results imply that chronic expression of polyQ-expanded AR, i.e. during development and prepuberty, is key to induce the full SBMA muscle pathology observed in patients. Our data support a model whereby chronic expression of polyQ-expanded AR triggers muscle atrophy through toxic (neomorphic) gain of function mechanisms distinct from normal (hypermorphic) gain of function mechanisms.

## 1. Introduction

Spinal and bulbar muscular atrophy (SBMA), also known as Kennedy’s disease, is an X-linked neuromuscular disease characterized by the selective dysfunction and degeneration of brainstem and spinal cord motor neurons (MNs) [1]. In addition, patients present with a wide array of peripheral organ and tissue dysfunction, including skeletal muscle weakness, wasting and atrophy [2,3]. SBMA is caused by exonic expansions of a CAG tandem repeat, resulting in an aberrantly elongated polyglutamine (polyQ) tract in the androgen receptor (AR) gene [4]. In normal individuals, the polyQ tract length ranges from 5 to 36 repeats and expansions over 38 repeats cause disease. The family of polyQ diseases also includes Huntington’s disease (HD), dentatorubral-pallidoluysian atrophy and six types of spinocerebellar ataxia (SCA) [5]. SBMA is triggered by the binding of polyQ-expanded AR to androgens and this is the reason why the disease fully manifests in males [6,7]. Females present with very mild symptoms even when they are homozygous for the mutation [8]. The sex-specificity of SBMA is well recapitulated in mouse models of disease [7,9,10]. SBMA is mainly associated with toxic gain functions (neomorphic GOF) of polyQ-expanded AR. In addition, patients show mild signs of androgen insensitivity and endocrine abnormalities, thereby implicating that the loss of AR function (LOF) also contributes to disease [11]. Although the androgen-dependent nature of the disease and experimental evidence support chemical and physical castration as a therapeutic strategy for SBMA [6,7,10], clinical trials based on this approach showed benefits only in a subset of patients [12]. Moreover, chronic androgen ablation may exacerbate the loss of AR function and muscle atrophy in patients. At the cellular level, polyQ expansions in the AR result in micro-aggregate and inclusion body formation. AR-positive inclusion bodies were detected in MNs in SBMA patients [13,14], as well as in pheochromocytoma (PC12) cells expressing an AR with 112 glutamine residues in response to androgen treatment [15] and agents capable of promoting inclusion body formation showed protective effects in cell and fly models of SBMA [16]. Since AR structure and function are well characterized [17], SBMA offers the opportunity to investigate the mechanistic relationship between structure/function and hypermorphic vs. neomorphic GOF of polyQ-expanded AR. 

AR is widely expressed in neuronal and non-neuronal cells. PolyQ expansions in the AR lead to selective lower MN degeneration through cell-autonomous mechanisms, which in turn results in skeletal muscle atrophy through non-cell-autonomous mechanisms [18]. Recent evidence supports the idea that polyQ-expanded AR also primarily affects skeletal muscle [19,20]. In addition, symptoms involving other peripheral organs, such as metabolic syndrome, sexual dysfunction and cardiac problems, further corroborate the possibility that polyQ-expanded AR is toxic not only to MNs, but also to other types of neurons and non-neuronal cells [21,22]. Genetic and pharmacologic activation of signaling pathways to induce polyQ-expanded AR degradation [23,24,25], or to silence its expression in muscle [19,20] have indeed provided evidence that expression of the disease protein in peripheral tissues is more than a bystander in disease pathogenesis. A unifying symptom in SBMA patients is a slow, though progressive, loss of the ability to walk and rise [22,26]. Muscle weakness and motor dysfunction are features invariably recapitulated in animal models of SBMA [27]. However, a clear mechanism explaining how polyQ expansions in the AR affect SBMA muscle is still missing. 

To gain insights into the mechanism through which polyQ-expanded AR causes disease, we generated mice for constitutive and inducible expression of the disease protein. We showed that expression of polyQ-expanded AR causes premature death, motor dysfunction, muscle atrophy, metabolic alterations and late-onset denervation. Importantly, polyQ expansion in the AR resulted in its accumulation in myonuclei in the form of aggregates and inclusion bodies in skeletal muscle. Expression of the mutant protein in adult mice was sufficient to recapitulate several, but not all disease manifestations in mice, implying additional toxic effects caused by chronic expression of the mutant protein possibly during development and prepuberty. Although overexpression of non-expanded AR caused muscle atrophy, it did not induce denervation, metabolic alterations in muscle, motor dysfunction and inclusion body formation. These observations showed that the SBMA transgenic mice with constitutive expression of mutant AR recapitulate the main features of disease manifestations previously described in other SBMA mice [7,9,10,19,27], thereby validating our newly generated mouse model of SBMA. Comparison of the phenotype of the SBMA mice with mice overexpressing non-expanded AR showed that overexpression of non-expanded AR causes muscle atrophy through hypermorphic GOF mechanisms and polyQ expansion causes specific alterations through neomorphic GOF mechanisms, which ultimately leads to dysfunction of the motor unit. 

## 2. Materials and Methods

### 2.1. Animals

Animal care and experimental procedures were conducted in accordance with the Italian Institute of Technology, the University of Trento and the University of Padova ethics committees and were approved by the Italian Ministry of Health. Mice were housed in filtered cages in a temperature-controlled room with a 12-hour light/dark cycle with *ad libitum* access to water and food. By random insertion transgenic mice were generated to express full-length hAR with a non-expanded polyQ tract with 24 glutamine residues (AR24Q) and a pathogenic polyQ tract (AR100Q) under the control of the cytomegalovirus immediate-early enhancer and the chicken beta-actin (pCAGGS) promoter. For conditional expression, hAR transgene expression was driven by the tetracycline-responsive element (TRE). Mice were genotyped by PCR on tail DNA using REDExtract-N-Amp Tissue PCR kit (Sigma-Aldrich, St. Louis, MO, USA) according to the manufacturer’s instructions. Primers for genotyping AR24Q and AR100Q mice: Forward 5’-CTTCTG GCGTGTGACCGGCG, reverse 5’-TGAGCTTGGCTGAATCTTCC. Primers for genotyping iAR100Q mice: Forward 5’-CGTATGTCGAGGTAGGCGTG, reverse 5’-TGAGCTTGGCTGAATCTTCC. Transgenic lines were backcrossed to the C57Bl6J background for more than 10 generations before subsequent analysis of phenotype and pathology. rtTA mice (Stock n.: 003273) were purchased from The Jackson Laboratory. iAR100Q/rtTA and controls mice were treated with doxycycline (Sigma Aldrich) in drinking water containing 5% sucrose (Sigma-Aldrich S8501). Motor function was assessed as previously shown [23,24,25,28]. Briefly, to perform the hanging wire test, the mouse was placed on a grid. The grid was shaken slightly three times to cause the mouse to grip the wires and then the grid was turned upside down. The latency to fall was recorded for a maximum time of 60 seconds. For grip strength (Bioseb, Vitrolles, France), rotarod (Ugo Basile, Varese, Italy) and hanging wire analyses mice were randomized and the operator was blind for genotype. Mice were sacrificed for humane endpoint when the mouse either lost 20% of body weight (BW) or showed the inability to move and signs of dehydration and cachexia.

### 2.2. Gene Copy Number Determination

Gene copy number (GCN) was evaluated using RT-qPCR using the qbase+ software [29]. Genomic DNA was extracted from 5 mice of each line using ReliaPrep™ gDNA Tissue Miniprep System (Promega, Madison, WI, USA). Specific primers were used to amplify the human AR transgene (hAR) and two housekeeping genes with a known GCN = 2: Gusb (Gusb glucuronidase beta) and Tfr (transferrin receptor). A mouse sample was used as an inter-run control and the transgenic mice were compared to the knock-in mouse gDNA, in which the first exon of the *AR* gene was substituted with polyQ expanded human allele, therefore resulting in GCN = 1 for hAR. Primers were designed as followed: hAR Forward 5’- CTTCACCGCACCTGATGTG, hAR Reverse 5’- TAAGGTCCGGAGTAGCTATC, mmuGusb Forward 5’- CCTGGATGTCCGGGTGAATC, mmuGusb Reverse 5’- GAGGTATGTGCACCGGGATG, mmuTfR Forward 5’- ATACCCCAAA TTTTGACCAGCC, mmuTfR Reverse 5’- GACCTTGCCTCAAAGAAAAACCT.

### 2.3. Histological Analysis

For muscle histology, tissues were flash-frozen in liquid nitrogen and embedded in optimal cutting temperature (OCT) compound (Tissue Tek, Sakura, Mestre, Italy). Cross-sections (10 μm thick) were cut with a cryostat (CM1850 UV, Leica Microsystems, Wetzlar, Germany) and processed for hematoxylin and eosin (H/E) and nicotinamide adenine dinucleotide (NADH) diaphorase staining, as previously described [23]. Images were taken using an upright epifluorescence microscope (Axio Imager M2, Zeiss, Oberkochen, Germany) equipped with an X-Cite 120Q fluorescence light source and a Zeiss Mrm Color Camera. Multichannel images and mosaics were taken using Zeiss Axio Vision Software (V.4.8.2 SP3). For Nissl staining of MNs in brainstem and spinal cord, deeply anesthetized mice were transcardially perfused with 4% paraformaldehyde (PFA) and tissues were collected, post-fixed in PFA at 4 °C for 16 h and then washed abundantly in phosphate buffered saline (PBS). Tissues were cut and mounted on slices. The samples were gradually dehydrated (70, 95 and 100% EtOH), stained with a solution containing 0.1% cresyl violet for 5 min, washed in water, gradually dehydrated (70, 95 and 100% EtOH), cleared in xylene and mounted in Eukitt (Bio Optica, Milano, Italy). Images were acquired with the inverted Zeiss Observer Z1 microscope. Toluidine blue was performed on sciatic nerve fixed in phosphate buffer 2% glutaraldehyde (cat. 07710, Polyscience, Warrington, PA, USA) to assess the number of axons and the myelin thickness on serial 1 μm thick nerve sections. Images were taken with a 100× objective on a Leica DM5000 microscope. Analysis of the number of axons and the g-ratio was performed using Image J and the G-Ratio Calculator plugin (http://cifweb.unil.ch). 

### 2.4. Biochemical Analysis

Tissues were snap-frozen in isopentane precooled in liquid nitrogen, stored at −80 °C, pulverized using pestle and mortar and homogenized in RIPA buffer containing protease inhibitors (Sigma), with either 0.1% or 2% sodium dodecyl sulfate (SDS). Lysates from muscles and liver were homogenized (homogenizer RZR 2052 control, Heidolph, Schwabach, Germany) at 600 rpm (20 times), sonicated and centrifuged at 15,000 rpm for 15 min at room temperature. Other tissue lysates were directly sonicated and centrifuged at 15,000 rpm for 15 min at room temperature. Protein concentration was measured using the bicinchoninic acid (BCA) assay method (Pierce, Thermofisher Scientific, Walthman, MA, USA). For Western blotting analysis, equal amounts of protein were separated in 7.5% Tris-HCl SDS-PAGE. Gels were blotted overnight onto 0.45-mm nitrocellulose membranes (162-0115, Bio-Rad, Hercules, CA, USA). Filter retardation assay was carried out as previously described [23]. For subcellular fractionation, samples were weighed and diluted 1:10 in Buffer A (20 mM HEPES, 100 mM KCl, 1 mM EDTA, 2 mM β-mercaptoethanol, 0.3% BSA), homogenized using a glass pestle in a glass potter and centrifuged at 800g for 10 min at 4 °C. The resulting supernatant (S1) was transferred to a new tube and the pellet was diluted 1:20 in Buffer A and centrifuged at 800 *g* for 10 min at 4 °C. The resulting supernatants (S1 + S2) were pooled and centrifuged at 3220 *g* for 5 min at 4 °C to pellet the nuclear fraction. The supernatant was recovered and centrifuged at 10000 *g* for 10 min at 4 °C to collect the cytosolic fraction. The following primary antibodies were used: AR (H280 sc-13062, 1:1000), β-tubulin (T7816, 1:5000), p62 (P0067, 1:1000), LC3B (L7543, 1:1000), Calnexin (ADI-SPA-860, 1:5000), phospho-Ser240/244-S6 (2215, 1:1000), total S6 (2317, 1:1000). Protein signals were detected using either the Odyssey Infrared Imaging System (Li-Cor, Lincoln, NE, USA) or the Alliance Q9 Mini chemidoc system (Uvitec, Cambridge, UK) with appropriate secondary antibodies. Quantifications were performed using ImageJ 1.45 software. Mitochondrial membrane potential was measured in isolated fibers from FDB by epifluorescence microscopy based on the accumulation of tetramethylrhodamine methyl ester (TMRM) fluorescence, as previously described [28]. 

### 2.5. Immunofluorescence Analysis

Muscles were isolated and immediately fixed in 4% paraformaldehyde (PFA). Bladder was used as whole-mount preparation, whereas gastrocnemius and flexor digitorum brevis (FDB) were further dissected into muscle bundles of about 20 myofibers. Samples were quenched in 50 mM NH_4_Cl for 30 min at room temperature and then saturated for 2 h in blocking solution (15% *vol*/*vol* goat serum, 2% *wt*/*vol* BSA, 0.25% *wt*/*vol* gelatin and 0.2% *wt*/*vol* glycine in PBS containing 0.5% Triton X-100). Primary antibodies were diluted in the same solution and then added to muscles for at least 48 h at 4 °C under gentle agitation. Antibodies: Anti-syntaxin-1A/1B (rabbit polyclonal, 1:200), anti-AR (H280 sc-13062, 1:200, Santa Cruz, Dallas, TX, USA), p62 (03-GP62-C1:200 ARP, Cambridge, UK). Muscles were extensively washed with PBS for at least 2 h and then incubated for additional 2 h with appropriate secondary antibodies (Life Technologies, Carlsbad, CA) supplemented with fluorescently labeled α-bungarotoxin (Life Technologies) to stain nicotinic acetylcholine receptors and with Hoechst 34580 (Sigma Aldrich). After extensive washes with PBS, muscles were rinsed with deionized water and mounted on a coverslip with mounting solution (S3023, Dako, Agilent, Santa Clara, CA) for microscopy examination. Images were collected with an epifluorescence microscope (Leica CTR6000) equipped with x5 N PL AN 0.12, x20 N PL AN 0.40 objectives or by an SP5 confocal microscope (Leica Microsystems) equipped with x100 HCX PL APO NA 1.4 objective or x40 HCX PL APO NA 0.75 or x63 PL APO NA 0.60. Fibre typing was analyzed in 10 µm quadriceps cryosections by immunofluorescence using combinations of the following monoclonal antibodies distributed by DSHB: BA-D5 (MyHC-I; 1:300), BF-F3(MyHC-IIb; 1:300), SC-71(MyHC-IIa; 1:300) and not stained myofibers were MyHC-IIx. Images were captured using a Leica DFC300-FX digital charge-coupled device camera by using Leica DC Viewer software and morphometric analyses were made using ImageJ. Images for the same analysis were collected with the same parameters of intensity and exposure.

### 2.6. Quantitative Real-Time PCR Analysis

Total RNA was extracted with TRIzol (Thermo Fisher Scientific, Walthman, MA, USA) and RNA was reverse-transcribed using the iScript Reverse Transcription Supermix (1708841 Bio-Rad) following the manufacturer’s instructions. Gene expression was measured by RT-qPCR using the SsoAdvanced Universal Sybr green supermix (1725274, Bio-Rad) and the C1000 Touch Thermal Cycler–CFX96 Real-Time System (Bio-Rad). The list of specific primers (Eurofins, Bayern, Germany) is provided in Appendix A. Gene transcript levels were normalized to actin (Mm01333821).

### 2.7. Statistical Analysis

Normality was tested using the Jarque–Bera test. To compare normally distributed measures across groups, student’s two sample t-tests and one-way analysis of variance (ANOVA) tests followed by Newman-Keuls honest significant difference post-hoc tests were used for two and more than two-group comparisons, respectively. When data did not follow a Gaussian distribution, we used the Mann-Whitney test to compare group differences. To evaluate BW and behavioral differences across genotype groups over time, Mann-Whitney tests between genotypes at each time point were used. Statistical comparison of Kaplan Meier survival curves was performed using the log-rank test. For all tests, the significance threshold was set at *p* < 0.05.

## 3. Results

### 3.1. PolyQ-Expanded AR, but not Non-Expanded AR, Causes Motor Dysfunction

To elucidate the molecular mechanisms underlying SBMA pathogenesis, we generated transgenic mice constitutively expressing human AR transgene (*hAR*) with 100 glutamine residues (AR100Q). As control for overexpression, we generated mice expressing AR24Q. GCN of the *hAR* transgene in AR24Q and AR100Q mice was estimated to be 77 ± 11 and 73 ± 14, respectively (Figure 1A). Male AR24Q mice suffered from severe hemorrhoids and developed difficulties to move in the cage, which required premature sacrifice, thereby resulting in a significantly reduced survival with a median of 26 weeks (χ2 Log−Rank = 16.03, *p* < 0.0001 relative to WT) (Figure 1B). 

Female AR24Q mice did not show any overt phenotype (Appendix A). Male AR100Q mice showed signs of weakness, atrophy and kyphosis and had a significantly reduced survival with a median of 13 weeks (χ2 Log−Rank = 17.1, *p* < 0.0001 relative to WT) (Figure 1B and Appendix A). Female AR100Q mice showed a significantly reduced survival with a median of 23 weeks (χ2 Log-Rank = 26.62, *p* < 0.0001 relative to WT) and presented a phenotype similar to that of male mice at the time of sacrifice (Appendix A). AR24Q mice were smaller compared to WT siblings starting from week 7 (Mann-Whitney test, *p* = 0.006), yet their BW progressively increased throughout their life (Figure 1C). Conversely, the BW of AR100Q mice was significantly lower compared to that of WT and AR24Q mice starting from week 7 (Mann-Whitney test, *p* = 0.006) and week 11 (Mann-Whitney test, *p* = 0.02) of age, re spectively and progressively decreased throughout their life span. We analyzed motor coordination by rotarod task and muscle force by hanging wire task (Figure 1D). Overexpression of non-expanded AR affected neither motor coordination nor muscle force. By rotarod task, AR100Q mice showed a progressive deterioration of motor coordination compared to control mice starting from week 10 (Mann-Whitney test, *p* = 0.006). By hanging wire task, AR100Q performance was different from that of control mice by 8 weeks of age (Mann-Whitney test, *p* = 0.04) and it progressively decreased throughout life span. These results indicate that polyQ-expanded AR and not non-expanded AR, causes motor dysfunction in vivo.

### 3.2. Denervation is a Late Event in SBMA Muscle

To determine whether alterations of motor function in AR100Q result from abnormalities in the motor unit, we analyzed MN number and soma area, axon, neuromuscular junction (NMJ) and skeletal muscle pathology. 

Consistent with previous findings [9,19,23], no changes were detected by Nissl staining in the number and soma area of MNs in transversal lumbar spinal cord sections of 12-week-old AR100Q mice as well as age-matched WT and AR24Q mice (Figure 2A). By toluidine blue staining of semi-thin sciatic nerve transversal sections, we did not find any gross abnormalities in both the number of axons and the g-ratio (axon diameter/nerve diameter) in 8-week-old AR100Q and control mice, indicating that myelin thickness and axon diameter are preserved (Figure 2B and Appendix A). Analysis of at least 50 NMJs from 3 mice/genotype revealed 100% one-to-one matching between acetylcholine receptor (AchR) clusters and the nerve terminals, stained with α-Bungarotoxin and the presynaptic marker syntaxin-1A/1B, respectively, in gastrocnemius, quadriceps, tibialis anterior (TA), extensor digitorum longus (EDL) and soleus of 8-week-old AR100Q and control mice (Figure 2C and data not shown). On the other hand, NMJ sinuous pretzel-like shape assumed a more fragmented pattern in 12-week-old AR100Q mice, revealing late-onset signs of degeneration of the NMJs (Figure 2D).

Based on these observations, we assessed whether AR100Q mice develop signs of denervation and myopathy at early (4 weeks), middle (8 weeks) and late stage (12 weeks for AR100Q mice and 24 weeks for AR24Q mice) of disease. We measured the transcript levels of embryonic myosin heavy chain 3 (*Myh3*), perinatal myosin heavy chain 8 (*Myh8*), myogenin (*MyoG*), muscle-specific kinase (*Musk*), acetylcholine receptor gamma (*Achrg*) and neural cell adhesion molecule (*Ncam*) (Figure 3A). We found a strong upregulation of the transcript levels of these genes starting from 8 weeks of age in AR100Q mice, which was exacerbated by 12 weeks of age. On the other hand, we did not find any significant upregulation of the transcript levels of these genes in AR24Q mice, except for *Musk*. Next, we analyzed muscle pathology by H/E staining (Figure 3B). In AR100Q mice at the late stage (12 weeks) of disease, severe muscle pathology was manifest and included a high number of atrophic, angulated and grouped fibers, together with large hypertrophic fibers with central nuclei, which is a feature of SBMA muscle [30] and a modest increase of perimysium connective tissue. Overall, pathology revealed a picture of denervation with myopathic signs that recapitulates the muscle pathology described in SBMA patients [31]. These observations indicate that denervation is a late event in this mouse model of SBMA, similarly to knock-in SBMA mice [9,28].

### 3.3. PolyQ-Expanded AR Alters Skeletal Muscle Homeostasis and Metabolism and Causes Mitochondrial Dysfunction

Next, we further analyzed skeletal muscle pathology. The weight of quadriceps and gastrocnemius muscles was significantly decreased by 16% and 12%, respectively, in 4-week-old AR24Q mice (Figure 4A). On the other hand, the weight of quadriceps, gastrocnemius, TA and EDL, but not soleus, was decreased by 25-through-50% in 8-week-old AR24Q and AR100Q mice, indicating that alteration of androgen signaling causes muscle atrophy in older animals. The SBMA muscle is characterized by an age-dependent glycolytic-to-oxidative fiber-type switch [3,28,30]. By NADH staining, the number of glycolytic fibers was decreased by 26% selectively in 8-week-old AR100Q mice and no changes were detected in WT and AR24Q mice (Figure 4B and Appendix A). Analysis of the cross-sectional area (CSA) of glycolytic and oxidative fibers revealed that the median CSA of glycolytic fibers was decreased by 35% and 10% in 4-week-old AR24Q and AR100Q mice, respectively and by 54% and 48% at 8 weeks of age, whereas the median CSA of oxidative fibers was decreased by 23% and 8% in 4-week-old AR24Q and AR100Q mice and by 5% and 11% at 8 weeks of age, respectively. By immunofluorescence analysis of myosin heavy chain (MyHC) subtypes, we found that the percentage of fiber-type in 8-week-old WT, AR24Q and AR100Q mice was 1%, 3% and 4% for type I fibers, 8%, 11% and 18% for type IIa fibers, 12%, 45% and 25% for type IIx fibers and 78%, 41% and 52% for type IIb fibers, respectively (Figure 4C). This indicates a progressive shift from type IIb fibers towards type IIx and IIa fibers in the quadriceps of AR24Q and AR100Q mice, respectively. Moreover, we found decreased expression of chloride channel 1 (Clcn1) in the muscle of SBMA mice compared to control mice, as previously reported in knock-in SBMA mice [9], suggesting altered muscle excitability (Figure 4D). 

We have previously shown that pathological processes occurring in SBMA muscle involve activation of anabolic and catabolic pathways [25,28,32]. Anabolic pathways that promote fiber hypertrophy result in activation via phosphorylation of protein kinase B (Akt) and mechanistic target of rapamycin (mTOR), which leads to phosphorylation and activation of p70 S6 kinase (p70S6K) that in turn phosphorylates the ribosomal protein S6 [33].

The morphological and functional changes described above were associated with increased phosphorylation of S6 in the muscle of 4- and 8-week-old AR24Q and AR100Q mice, suggesting sustained activation of anabolic pathways (Figure 5A). On the other hand, we found that the transcript levels of genes involved in protein degradation via the ubiquitin-proteasome system (UPS) and autophagy, such as the E3 ubiquitin ligases, specific of muscle atrophy and regulated by transcription (*Smart*) and muscle ubiquitin ligase of the SCF complex in atrophy-1 (*Musa1*), forkhead box O 3a (*Foxo3a*), microtubule-associated protein 1A/1B-light chain 3 (*LC3*) and sequestosome 1 (*Sqstm1,* p62) were increased specifically in 8-week-old AR100Q mice, suggesting enhanced UPS and autophagy activity specifically in SBMA mice (Figure 5B). By Western blotting, we found a significant accumulation of LC3II, suggesting an increased number of autophagosomes and p62, which although not significant suggests a block of the autophagy flux, as previously reported in other SBMA mouse models and patients (Figure 5C) [28,34,35]. These observations show that the expression of polyQ-expanded AR results in concurrent enhancement of hypertrophy and atrophy pathways in muscle, as previously observed in knock-in mouse models of SBMA and patients [28,30]. Notably, activation of these biochemical markers of muscle atrophy was not detected in AR24Q mice, further implying different mechanisms underlying the muscle pathology of AR24Q and AR100Q mice.

Activation of autophagy can be caused by or associated with mitochondrial dysfunction [36]. Mitochondrial abnormalities have been extensively described in SBMA cells, mice and patients [30,37]. By measuring mitochondrial membrane potential in myofibers isolated from the FDB muscle upon treatment with the F_1_F_0_-ATPase blocker, oligomycin, we found that the number of fibers with mitochondria depolarized by oligomycin was significantly (*p* < 0.05) increased in 8- and not 4-week-old AR100Q mice as well as control mice (Figure 5D and Appendix A). PGC1α (*Ppargc1a*) promotes mitochondrial biogenesis [38]. By real-time PCR analysis, *Ppargc1a* transcript levels were upregulated in the quadriceps of 4-week-old AR24Q mice and downregulated in AR100Q mice (Figure 5E). The mitochondrial fission marker, dynamin related protein 1 (*Dnm1l*, Drp1), was upregulated by 2-fold in AR100Q mice starting from 4 weeks of age, whereas the mitochondrial fusion marker, opticatrophy gene 1 (*Opa1*), did not change. Collectively, these observations indicate that AR100Q mice show severe and progressive muscle atrophy, altered muscle homeostasis and metabolism and mitochondrial dysfunction.

### 3.4. PolyQ Expansion Leads to 2% SDS-Resistant Aggregate and Inclusion Body Formation Selectively in Skeletal Muscle

Next, we analyzed AR expression in vulnerable tissues. By Western blotting analysis of spinal cord, brainstem and skeletal muscle total lysates monomeric AR was increased by 5-, 15- and 48-fold, respectively, in 8-week-old AR24Q mice, whereas monomeric polyQ-expanded hAR was increased by 1-, 1.4- and 1.5-fold in AR100Q mice, respectively (Figure 6A). PolyQ-expanded proteins and misfolded proteins linked to MN diseases, such as mutant superoxide dismutase 1, form micro-aggregates, which can be detected as high-molecular-weight (HMW) species that accumulate in the stacking portion of polyacrylamide gels and by filter retardation assay [23,39,40]. These micro-aggregates can be detected in the presence of both low (0.1%) and high (2%) concentrations of SDS and in the latter case such species are defined as 2% SDS-resistant aggregates [41]. No 2% SDS-resistant aggregates were detected in both spinal cord and brainstem, in which we rather detected polyQ-expanded AR aggregates in the milder buffer containing 0.1% SDS (Figure 6A, Appendix A), as previously reported in other transgenic SBMA mouse models with constitutive expression of AR [7,23]. Rather, 2% SDS-resistant aggregates of polyQ-expanded AR were detected in skeletal muscle (Figure 6A,B). AR aggregation has been extensively characterized in neurons, but not in muscle [7,15,42,43,44]. Therefore, we further explored this aspect of muscle pathology. PolyQ-expanded AR aggregation was detected as early as 4 weeks of age in muscles composed of both fast-glycolytic and slow-oxidative fibers and that degenerate in SBMA, such as quadriceps, gastrocnemius and TA, in muscles mainly composed of fast-glycolytic fibers, such as EDL and to a lower extent in muscles mainly composed of slow-oxidative fibers, such as soleus and diaphragm and heart that are mildly affected in SBMA, indicating that polyQ-expanded AR aggregation occurs early and is affected by muscle metabolism (Figure 6C). Importantly, no polyQ-expanded AR aggregation was detected in other peripheral tissues, including testis and adrenal glands, which express levels of AR similar to muscle (Appendix A). Moreover, the amount of aggregated polyQ-expanded AR compared to monomeric AR was higher (1.4-fold) in male compared to female AR100Q mice (Appendix A). Notably, by performing subcellular fractionation on frozen muscles, we observed that polyQ-expanded AR aggregates were enriched in the nuclear fraction (Figure 6D). These results show that deposition of 2% SDS-resistant AR aggregates is polyQ length-dependent, is at least in part androgen-dependent and mainly occurs in skeletal muscle myofibers as an early event. 

Next, we asked whether polyQ-AR forms inclusion bodies in vivo. To address this question, we performed immunohistochemical analysis of AR subcellular localization in intact fibers isolated from different types of muscles of AR100Q and control mice (Figure 6E,F and Appendix A). Non-expanded AR was homogeneously diffused in both the cytosolic and nuclear compartments of the myofibers of AR24Q mice. In the myofibers of AR100Q mice, polyQ-expanded AR showed a diffused distribution at 15 days of age, which became enriched in the nuclei by 4 weeks of age with a distinct punctate pattern by 8 weeks of age and lower levels of protein detected in the cytosol compared to control mice. Nuclear inclusions were detected also in female mice, yet to a lower extent compared to male mice (55% and 43% of nuclei with aggregates in males and females, respectively), suggesting that androgens enhance the process of AR-positive inclusion body formation. In striated muscles, inclusions were detected in all muscles analyzed here. Inclusions were detected also in cardiac muscle, yet to a much lower extent compared to skeletal muscles. No inclusions were detected in smooth muscles, such as bladder, despite high expression of polyQ-expanded AR. Inclusions were detected both in sub-synaptic and non-sub-synaptic nuclei (Appendix A).

By confocal microscopy analysis, at 15 days of age polyQ-expanded AR was enriched in the myonuclei, where it showed a diffused distribution similar to that of AR24Q (Figure 7A). By 4 weeks of age, when serum testosterone levels physiologically increase in male mice, nuclear enrichment augmented and was associated with the formation of inclusion bodies. Notably, we observed several small AR-positive puncta in the myonuclei at 4 weeks of age and less yet far larger inclusion bodies at 8 weeks of age. AR-positive inclusion bodies were also recognized by an antibody against p62, indicating that these mice develop p62-positive pathology in skeletal muscle (Figure 7B). These results indicate that in skeletal muscle polyQ expansion in AR results in nuclear enrichment with progressive deposition of the disease protein into intranuclear inclusions. The observation that aggregation and inclusion body formation preceded pathological processes occurring in muscle suggests a causative link between these phenomena and the morphological, functional and structural abnormalities detected in skeletal muscle.

### 3.5. Expression of PolyQ-Expanded AR in the Adulthood Elicits Some, but not All Aspects of Disease Manifestations in Mouse

The transgenic mice described above express *hAR* in a constitutive fashion, i.e., from development. Therefore, we asked whether the expression of mutant AR in the adult mouse recapitulates the major symptoms of SBMA observed in mice with constitutive expression of polyQ-expanded AR. To acutely induce polyQ-expanded AR expression in vivo, we generated transgenic mice for doxycycline (dox)-inducible expression of AR100Q (hereafter referred to as iAR100Q for inducibleAR100Q), which were subsequently crossed with mice expressing rtTA under the CMV promoter (hereafter referred to as rtTA mice) [45]. iAR100Q /rtTA double transgenic mice were fertile, viable and did not show any overt phenotype in the absence of dox. Administration of dox (0.1, 0.2 and 1g/L) to pregnant dams did not give birth to any double-positive pups, indicating that induction of *hAR* transgene is highly toxic for embryo development, possibly due to high copy number insertion (Appendix A and data not shown). To induce polyQ-expanded AR expression in adulthood, while avoiding compensatory or toxic effects occurring during development and before sexual maturation, we started dox treatment at 6 weeks of age (Figure 8A), which is approximately the time of disease onset in transgenic SBMA mice constitutively expressing mutant AR (Figure 1). While treatment of iAR100Q/rtTA mice with 0.1 and 0.2 g/L of dox was not sufficient to induce hAR expression and trigger disease manifestations, administration of 1 g/L of dox efficiently induced *hAR* expression, but not aggregation (Figure 8B). Dox treatment caused premature death (Log-rank test *p* = 0.001) with a median survival of 8 weeks of age (maximum life duration was 15 weeks of age) (Figure 8C). Similar to the transgenic mice constitutively expressing mutant AR, iAR100Q/rtTA mice manifested a progressive loss of BW and reduced muscle strength (Figure 8D,E), a phenotype not observed in WT, rtTA and iAR100Q mice (Appendix AA–D). 

Rotarod performance was not affected by Dox treatment mice (Appendix A). Dox treatment reduced quadriceps and not soleus, muscle mass by 21% in iAR100Q /rtTA mice, but not in WT, rtTA and iAR100Q mice (Figure 8F and Appendix A). By NADH and immunofluorescence analysis, we found that Dox treatment caused neither metabolic alterations nor fiber-type switch (Figure 8G and Appendix A). Rather, dox treatment resulted in a significant reduction of the mean CSA by 37% and 26% of glycolytic and oxidative fibers, respectively. Importantly, these observations indicate that post-natal induction of AR100Q expression is sufficient to cause premature death, loss of body weight as well as of muscle force and mass. However, two weeks of dox treatment were not sufficient to alter motor coordination and muscle metabolism and induce mutant AR aggregation and fiber-type switch, thereby implying that these components require a longer time window of polyQ-expanded AR expression to trigger this level of muscle pathology.

## 4. Discussion

A large body of evidence supports the idea that the skeletal muscle is a key site of toxicity in SBMA [19,20,28] and as such it represents a valuable target for therapy development [23,25,32,46]. 

Here we identified pathological processes associated with hypermorphic and neomorphic AR GOF in vivo. AR24Q and AR100Q mice showed atrophy of both glycolytic and oxidative fibers and a progressive fiber-type switch towards the slow fiber-type, which was exacerbated by polyQ expansion. This supports the idea that these aspects of muscle pathology are associated with the enhancement of AR native functions (hypermorphic GOF) with the contribution of toxic functions (neomorphic GOF). However, AR100Q muscles displayed unique features, like glycolytic-to-oxidative metabolic switch, aberrant expression of Clcn1, signs of denervation, induction of atrogenes and autophagy genes and mitochondrial pathology, which are all ascribable to toxic, polyQ-AR dependent, GOF mechanisms. The muscle of AR24Q mice was characterized by decreased mass starting from 1 month of age, when muscles are still developing and later on in adult mice. This observation suggests a growth delay or impairment upon overexpression of normal AR from development throughout adulthood. Importantly, lack of induction of bona fide biochemical markers of muscle atrophy, such as increased expression of atrogenes and autophagy genes, implies a different mechanism underlying muscle atrophy in AR24Q mice with respect to AR100Q mice and other models of muscle atrophy, such as fasting, denervation, neuromuscular diseases and myopathies. Further investigation may shed light onto the pathogenetic processes occurring in mice that model a hypermorphic AR GOF, thereby providing a molecular link between muscle atrophy and disorders with amplification of AR function. 

Myofibers are classified based on the pattern of MyHC expression, contractile kinetics, resistance to fatigue and metabolic properties [47,48]. Fast myofibers (type IIb) contract quickly and develop the greatest power, they consume a large amount of energy and they undergo fast-fatigue as they use glycolytic metabolism to produce energy. Although much weaker and slower, slow myofibers (type I) contract for longer periods without fatiguing as they use oxidative phosphorylation. A range of intermediate fiber types (I/IIa 
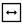
 IIa 
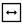
 IIa/IIx 
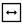
 IIx 
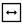
 IIx/IIb) is endowed with mixed properties. The more pronounced fiber-type switch in MyHC types may contribute to the loss of muscle force and performance of SBMA mice. A major determinant for the plastic remodeling of myosin isoforms in adult muscles is denervation [48]. In aging and ALS, fiber-type switch correlates with the progressive death of fast-fatigable MNs. Loss of fast-fatigable MNs is compensated, at least at initial stages, through reinnervation of orphan fibers by slow fatigue-resistant MNs, a process achieved via compensatory sprouting leading to muscle fiber-type grouping [49,50]. Notably, although we did not find physical denervation of the myofibers, NMJ morphology progressively deteriorated in SBMA mice, indicating NMJ instability at the late stage of disease. Overall, this scenario supports a functional, rather than a structural, impairment of the motor unit likely causing aberrant communication between muscle and MN, ultimately contributing to the loss of motor performance. Nonetheless, in AR100Q mice fiber-type changes preceded denervation. These observations imply that all these pathological processes may result from intrinsic processes occurring in muscle and in other peripheral tissues, the nature of which remains to be clarified

In skeletal muscle, we observed an age-dependent enrichment of polyQ-expanded AR in the nuclei of the myofibers. AR nuclear translocation is a key post-translational event that occurs in response to androgen binding and upon dissociation from heat shock proteins [17]. In AR100Q mice, the accumulation of AR in the myonuclei was observed very early and it preceded the appearance of symptoms. This phenomenon can result from either increased nuclear import or decreased nuclear export. AR nuclear import is mediated by the interaction with importin-α and importin-^®^ via its bipartite nuclear localization signal [51]. Upon transport through the nuclear pore complex inside the nucleus, the cargo protein is released from importin family proteins by the Ras family GTPase Ran. It is possible that polyQ expansions in the AR result in increased interaction with components of the nuclear pore complex, leading to aberrant accumulation of the disease protein in the nucleus. Accumulation of AR in the nucleus may also derive from malfunctions of the nucleocytoplasmic transport machinery, which is indeed defective in several neurodegenerative diseases [52,53]. Importantly, nuclear export of mutant AR was recently shown to be altered in SBMA in the absence of defects of the nucleocytoplasmic transport machinery [54]. Nuclear localization of mutant AR is necessary, albeit not sufficient, to cause disease [55,56]. Localization of polyQ-expanded AR, huntingtin, ataxin-1 and other proteins in the nucleus is toxic in vitro and in vivo [55,57,58,59]. Mutations that lead to cytosolic retention of polyQ-expanded AR attenuated toxicity in cells, flies and mice [16,55,56,57]. Therefore, aberrant accumulation of polyQ-expanded AR in the myonuclei may trigger or contribute to muscle atrophy, leading to motor dysfunction and premature death.

A key feature of muscle pathology of AR100Q mice was the presence of 2% SDS-resistant AR aggregates and intranuclear inclusion bodies. Inclusion body formation was an early event that correlated with disease progression and outcome in SBMA mice. During muscle development, polyQ-expanded AR had a diffuse distribution in the nuclei, which then transitioned to a punctate pattern as serum testosterone levels increased. Protein misfolding, aggregation and inclusion body formation are associated not only with toxic GOF, but also with LOF mechanisms. Several proteins of the protein quality control system, such as proteasome subunits, chaperones and ubiquitin, together with cytoskeletal proteins, transcription factors and RNA binding proteins, were detected in inclusions. Sequestration of these proteins may disrupt cellular proteostasis, leading to aggregation of metastable proteins, ultimately resulting in a disease state [60]. The proteostasis network is regulated by cell-autonomous and non-cell-autonomous signaling through communication between subcellular compartments and across different cells and tissues [61,62]. Disruption of proteostasis in muscle may then result in MN dysfunction and degeneration in SBMA as well as in other MN diseases. Inclusions may have a different impact on neurotoxicity depending on the subcellular compartment in which they form. For instance, the formation of a specific type of inclusion bodies known as aggresomes in the cytosol may be protective, as they represent sites of protein degradation [63,64]. Moreover, mutations that lead to cytosolic retention of polyQ-expanded AR may favor degradation of the disease protein via autophagy [56]. Whether aggregates and inclusions either positively contribute to the irreversible neurodegenerative process that leads to MN loss or are the result of a physiological defensive process that cells activate to confine toxic species to specific subcellular compartments remains a matter of debate. Neuronal intranuclear inclusion formation did not correlate with neuronal death in SBMA [65,66,67], HD [59] and SCA1 [58], indicating that this is a phenomenon distinct from neuronal death. Rather, neurons with inclusion bodies from Alzheimer’s disease and Parkinson’s disease patients seem to be healthier than neurons with no detectable inclusions [68,69]. Inclusion formation did not correlate with neurodegeneration in HD and polyQ-expanded huntingtin-positive inclusions were more abundant in the cortex than striatum [70]. Consistent with these observations, inclusion body formation correlated with a higher probability of survival of neurons expressing polyQ-expanded huntingtin in vitro [71] and in vivo [72], supporting the concept that inclusion bodies may represent a defense strategy of the cells against protein misfolding and aggregation. Intriguingly, the size of inclusion bodies increased with age in HD brain [73], as well as in SBMA myonuclei. Although we cannot establish whether these inclusions form as an adaptive or maladaptive response of the cells to the presence of polyQ-expanded AR, several observations lead us to speculate that these species are important for disease pathogenesis. These species were detected exclusively in the nuclei of myofibers, their formation correlated with the physiological increase of androgens in mouse and the extent of accumulation of polyQ-expanded AR into inclusions correlated with disease severity. 

Here, we generated transgenic mice for inducible expression of polyQ-expanded AR. Using this new model, we established that premature death, motor dysfunction, muscle weakness and atrophy can be observed when mutant AR is expressed in adulthood. Importantly, we did not observe mutant AR aggregation, metabolic alterations, fiber-type switch and altered motor coordination upon induction of expression of the disease protein for two weeks. These findings are consistent with the idea that expression of polyQ-expanded AR during development and before puberty is important, even if from this age on it may take years to elicit symptoms that ultimately lead to a diagnosis. Several questions remain to be addressed. We found that pathological events in muscle, such as metabolic alteration, fiber-type switch, mitochondrial dysfunction and activation of catabolic pathways, all precede denervation. Which cell-autonomous and non-cell-autonomous processes occur in muscle to elicit these signs of pathology? Is aggregation and inclusion body formation pathogenetic? Why are the mitochondria dysfunctional? Addressing these questions may provide fundamental knowledge onto the physiological role of AR in muscle and may lead to development of novel intervention for this yet incurable and untreatable neuromuscular disease.

## Figures and Tables

**Figure 1 cells-09-00325-f001:**
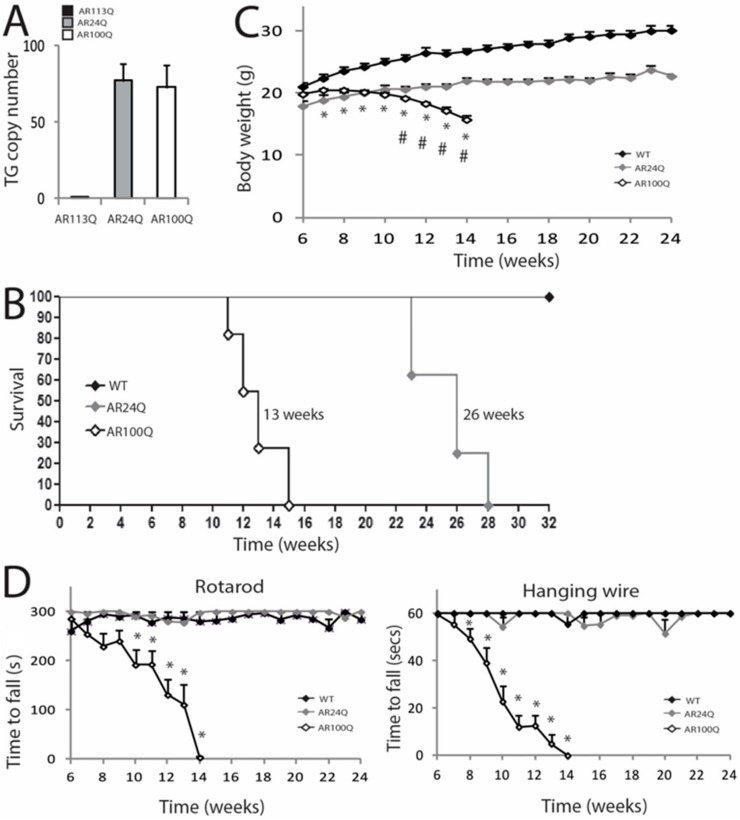
AR100Q transgenic mice show reduced life span and progressive motor dysfunction. (**A**) Analysis of transgene copy number in knock-in (AR113Q) and transgenic (AR24Q and AR100Q) male mice (*n* = 5). (**B**) Kaplan-Meier survival curves of WT (*n* = 8), AR24Q (*n* = 8) and AR100Q (*n* = 11) male mice. Survival curves were compared using Log-rank (Mantel-Cox) test. (**C**) Temporal changes in mean BW of WT (*n* = 8), AR24Q (*n* = 7) and AR100Q (*n* = 14) male mice. (**D**) Temporal changes in mean rotarod and hanging wire task performance in WT (*n* = 7), AR24Q (*n* = 5) and AR100Q (*n* = 12) male mice. Graphs, mean ± SEM. Statistical testing: Mann Whitney test was used to test the difference between genotypes in (**C**) and (**D**). In (**C**) * *p* < 0.01 for AR100Q vs WT; # *p* < 0.001 for AR100Q vs AR24Q. In (**D**) * *p* < 0.05.

**Figure 2 cells-09-00325-f002:**
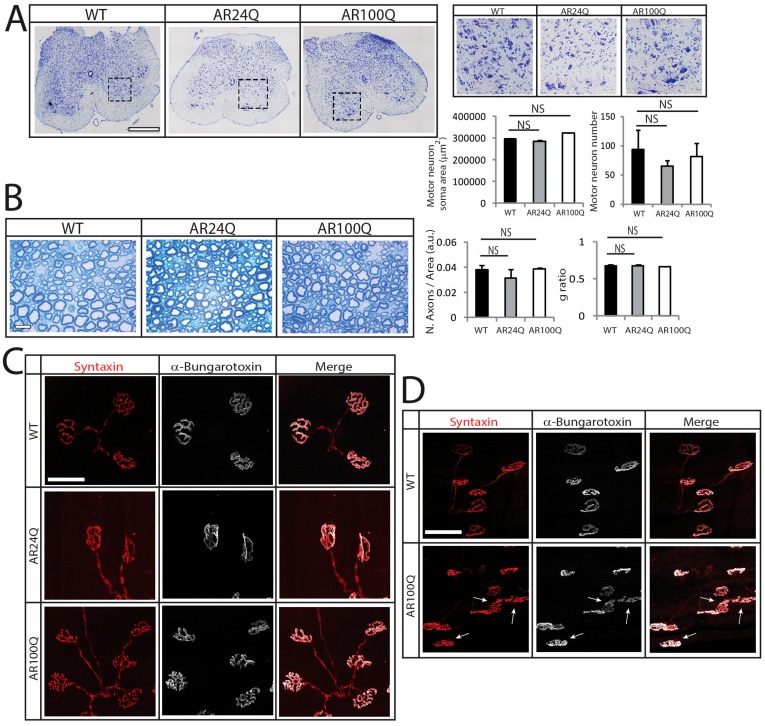
Late-onset alteration of NMJ morphology in AR100Q mice. (**A**) Nissl staining analysis of MN number and soma area in the lumbar spinal cord transversal sections of 12-week-old WT, AR24Q and AR100Q male mice (*n* = 3). Inset position is shown by the dashed square box. Bar, 500 micron. (**B**) Toluidine blue staining of semi-thin sciatic nerve transversal sections of 8-week-old WT, AR24Q and AR100Q male mice (*n* = 3). Bar, 10 micron. (**C**–**D**) Immunohistochemical analysis of NMJ pathology in the quadriceps muscle of 8-week-old (**C**) and 12-week-old (**D**) WT, AR24Q and AR100Q male mice (*n* = 3). Arrows indicate fragmented NMJs. Bar, 50 microns. Shown are representative images. Graphs, mean ± SEM. Statistical testing: One-way ANOVA followed by Newman-Keuls post-hoc test; NS, non-significant.

**Figure 3 cells-09-00325-f003:**
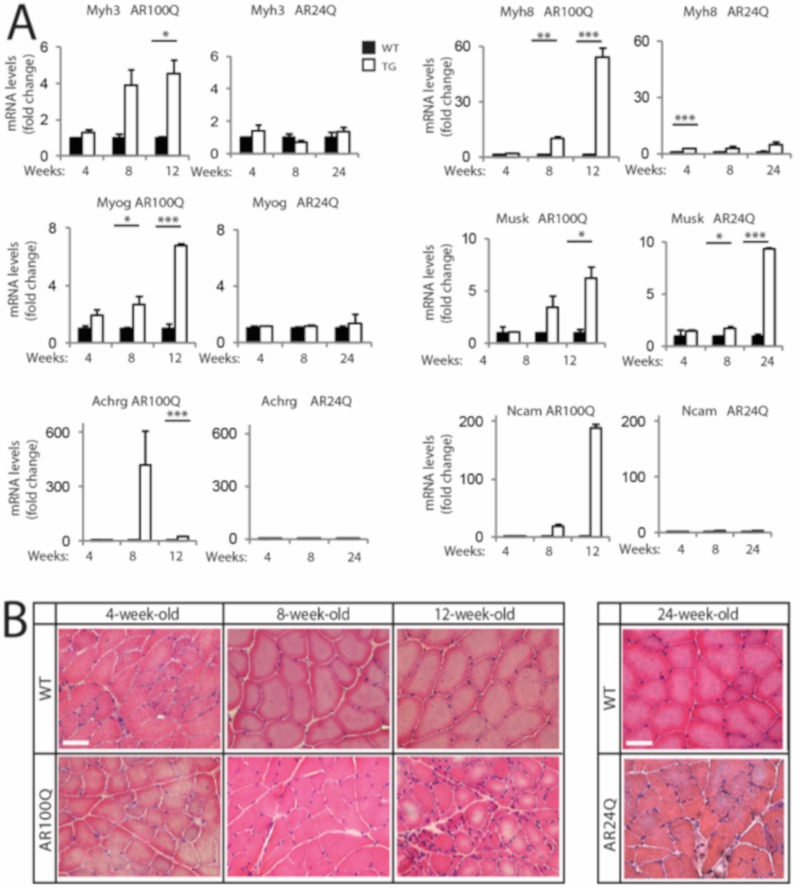
Signs of muscle atrophy at the late stage of disease in AR100Q mice and not AR24Q mice. (**A**) Real-time PCR analysis in the quadriceps of WT, AR24Q and AR100Q male mice (*n* = 3–5). (**B**) H/E staining of transversal sections of quadriceps muscle of WT, AR24Q and AR100Q male mice (*n* = 3). Bar, 50 microns. Shown are representative images. Graphs, mean ± SEM. Statistical testing: One-way ANOVA followed by Newman-Keuls post-hoc test; * *p* < 0.05, ** *p* < 0.01, *** *p* < 0.001.

**Figure 4 cells-09-00325-f004:**
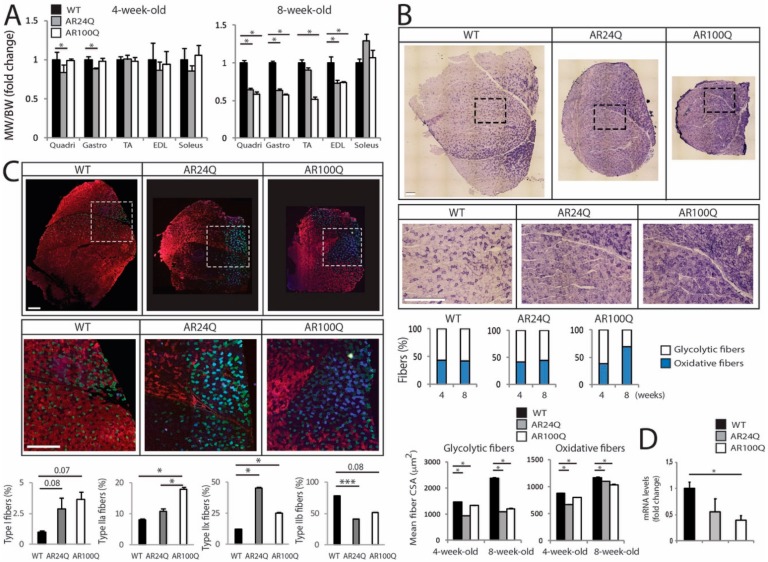
Switch from type IIb to type IIa and IIx fibers and altered muscle metabolism in AR100Q mice. (**A**) Muscle weight (MW) normalized to BW in 8-week-old WT (*n* = 14), AR24Q (*n* = 7) and AR100Q (*n* = 16) male mice. (**B**) NADH analysis of quadriceps and gastrocnemius of 4- and 8-week-old WT, AR24Q and AR100Q male mice (*n* = 3). Shown are representative images from 8-week-old mice. Inset position is shown by the dashed square box. Bars, 500 micron. Number of fibers (4-week-old): 1224 WT, 1224 AR24Q and 1237 AR100Q. Number of fibers (8-week-old): 1232 WT, 1257 AR24Q and 1219 AR100Q, all from 3 mice/genotype/age. (**C**) Immunofluorescence analysis of MyHC type I (blue), IIa (green), IIx (black) and IIb (red) from 8-week-old AR24Q and AR100Q male mice (*n* = 3). Shown are representative images. Inset position is shown by the dashed square box. Bars, 500 micron. (**D**) Real-time PCR analysis in the quadriceps of 8-week-old WT, AR24Q and AR100Q male mice (*n* = 3–4). Graphs, mean ± SEM. Statistical testing: One-way ANOVA followed by Newman-Keuls post-hoc test; * *p* < 0.05.

**Figure 5 cells-09-00325-f005:**
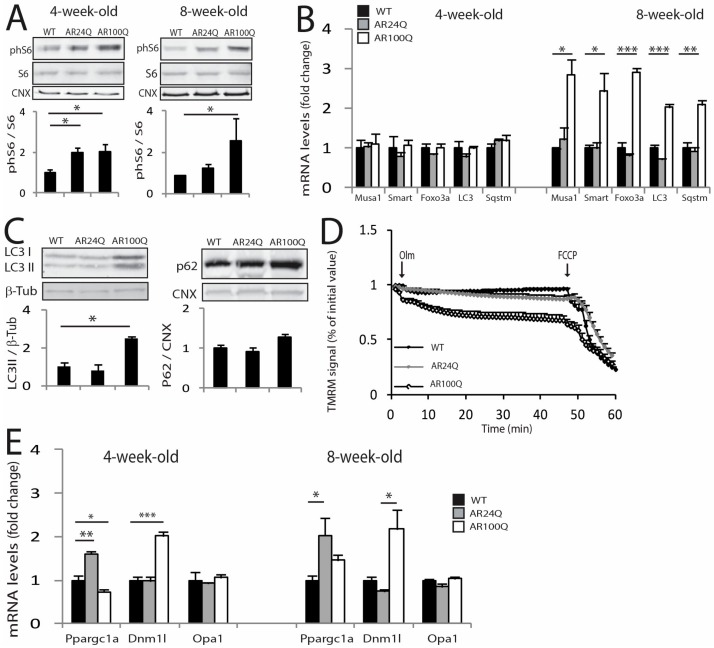
Activation of atrophy pathways and mitochondrial pathology selectively in the muscle of AR100Q mice. (**A**) Western blotting analysis of phosphorylated S6 (phS6) in the quadriceps of WT, AR24Q and AR100Q male mice (*n* = 3). phS6 was detected with a specific antibody that recognized S6 when phosphorylated at serines 240 and 244 and total S6 with a specific antibody. (**B**) Real-time PCR analysis of the transcript levels of the indicated genes in the quadriceps of WT, AR24Q and AR100Q male mice (*n* = 3). (**C**) Western blotting analysis of LC3I and II and p62 in the quadriceps of 8-week-old WT, AR24Q and AR100Q male mice (*n* = 3). (**D**) Mitochondrial membrane depolarization measured in fibers isolated from FDB muscle of 8-week-old WT, AR24Q and AR100Q male mice (*n* = 3). Olm, oligomycin; FCCP, protonophore carbonyl cyanide *p-*trifluoromethoxyphenylhydrazone TMRM, tetramethylrhodamine methyl ester. (**E**) Real-time PCR analysis, as described in (**B**). Graphs, mean ± sem. Statistical testing: (A, right panel) Pairwise Mann Whitney tests between genotypes; for all the other panels, one-way ANOVA followed by Newman-Keuls post-hoc test. * *p* < 0.05, ** *p* < 0.01, *** *p* < 0.001.

**Figure 6 cells-09-00325-f006:**
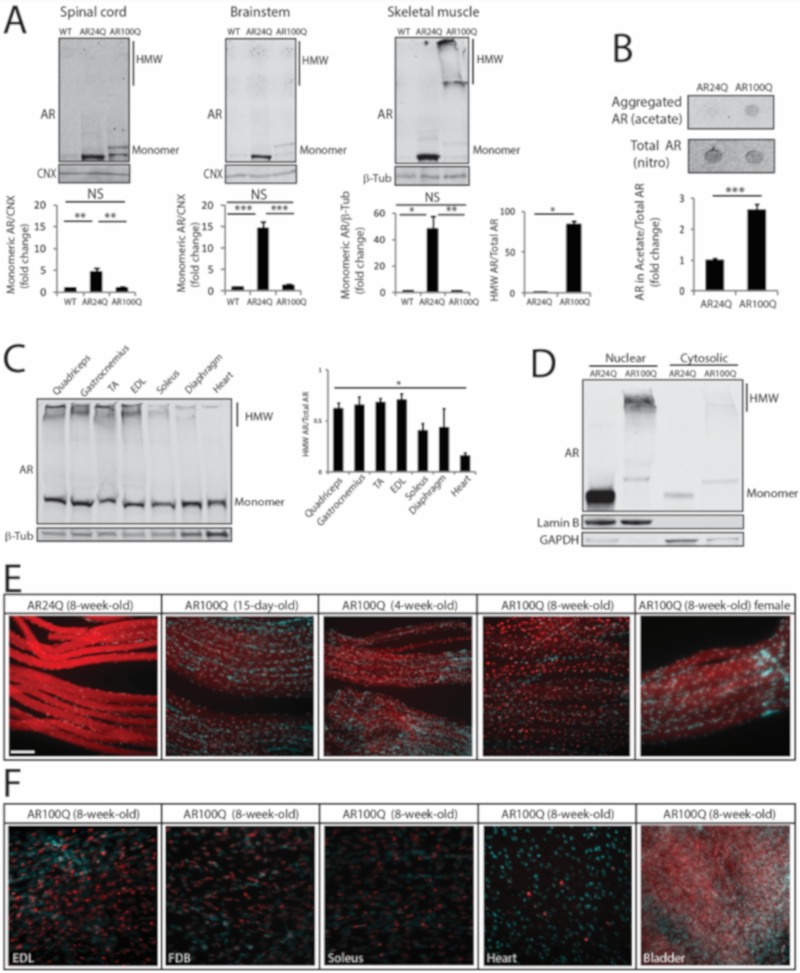
PolyQ-expanded AR forms 2% SDS-resistant aggregates and inclusion bodies in skeletal muscle. (**A**) Western blotting analysis of AR in 8-week-old WT, AR24Q and AR100Q male mice (*n* = 3). Quantification is shown at the bottom of each panel. (**B**) Filter retardation assay of total protein extracts from quadriceps muscle of 8-week-old AR24Q and AR100Q male mice (*n* = 3). (**C**) Western blotting analysis of AR in the indicated tissues from 4-week-old AR100Q male mice (*n* = 3). (**D**) Western blotting analysis of nuclear and cytosolic fractions from quadriceps muscle of 8-week-old AR24Q and AR100Q male mice (*n* = 2). (**E**,**F**) Immunofluorescence analysis of AR subcellular localization in intact fibers from gastrocnemius muscle (**E**) and the indicated muscles (**F**) of AR24Q and AR100Q male and female (where indicated) mice. Bar, 100 microns. In (**A**–**D**) AR was detected with a specific antibody and calnexin (CNX) and beta-tubulin (β-Tub) were used as loading controls. HMW, high-molecular-weight species. In (**D**) lamin B and GAPDH were used as loading controls of nuclear and cytosolic fractions. Graphs, mean ± sem. Statistical testing: Student’s t-test in (**A**, panel HMW species) and (**B**); one-way ANOVA followed by Newman-Keuls post-hoc test (**C**); * *p* < 0.05, ** *p* < 0.01, *** *p* < 0.001. NS, non-significant. In (**E**,**F**) AR (red) was detected with a specific antibody and nuclei with DAPI (blue) in intact fibers. Shown are representative images of at least 3 mice.

**Figure 7 cells-09-00325-f007:**
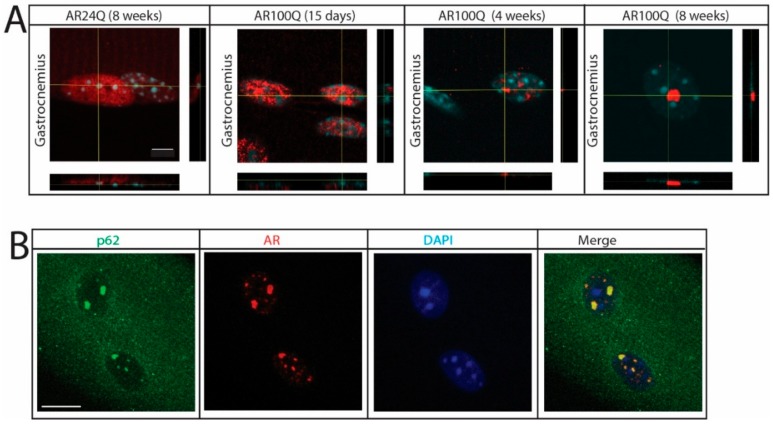
Nuclear enrichment and p62-positive pathology in SBMA muscle. (**A**) Confocal microscopy analysis of AR subcellular localization in intact myofibers from gastrocnemius muscle of AR24Q and AR100Q male mice. Bar, 5 micron. (**B**) Immunofluorescence analysis of AR and p62 subcellular localization in intact fibers from gastrocnemius muscle of 8-week-old AR100Q male mice. Bar, 10 micron. AR (red) and p62 (green) were detected with specific antibodies and nuclei with DAPI (blue). Shown are representative images of at least 3 mice.

**Figure 8 cells-09-00325-f008:**
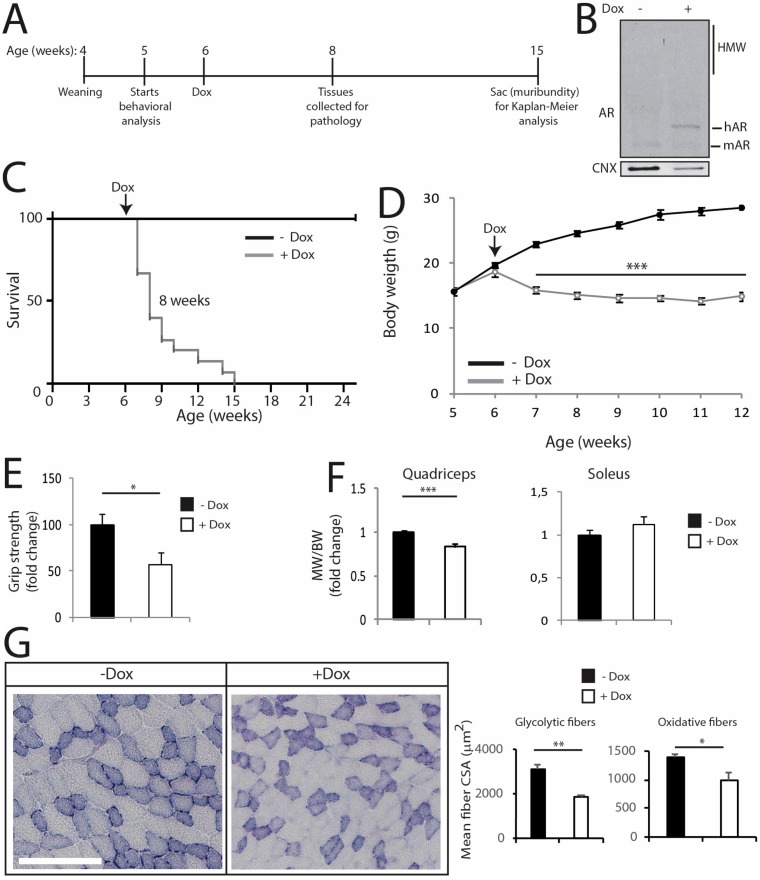
Induction of polyQ-expanded AR expression in the adulthood is sufficient to cause several, but not all disease manifestations in mouse. (**A**) Scheme of treatment of conditional SBMA mice with doxycycline (dox). (**B**) Western blotting analysis of AR expression in the quadriceps of 8-week-old iAR100Q/rtTA male mice treated with either vehicle (−) or dox (+). Shown is one experiment representative of three. (**C**) Kaplan-Meier analysis of survival of iAR100Q /rtTA male mice treated with either vehicle (*n* = 8) or dox (*n* = 15). Survival curves were compared using Log-rank (Mantel-Cox) test. (**D**) Temporal changes in mean BW of iAR100Q /rtTA male mice treated with either vehicle (*n* = 8) or dox (*n* = 15). (**E**) Grip strength analysis of muscle force of 8-week-old iAR100Q/rtTA male mice treated with either vehicle (*n* = 8) or dox (*n* = 8). (**F**) MW normalized to BW in 8-week-old iAR100Q/rtTA male mice treated with either vehicle (*n* = 4) or dox (*n* = 5). (**G**) Analysis of the mean CSA of fibers in the TA of 8-week-old iAR100Q/rtTA male mice (*n* = 3–4) treated with either vehicle or dox. Number of fibers: 1600 fibers vehicle, 1200 dox. Graphs, mean ± SEM. Statistical testing: (**D**) Mann-Whitney test for each time point; (**E**–**G**) Student’s t test; * *p* < 0.05, *** *p* < 0.001.

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
