# Peer review of "Polyglutamine-Expanded Androgen Receptor Alteration of Skeletal Muscle Homeostasis and Myonuclear Aggregation Are Affected by Sex, Age and Muscle Metabolism"

_cells, 2020, doi:10.3390/cells9020325_

Round 1

Reviewer 1 Report

This study describes generation of new mouse models for studying muscle pathology in SBMA - spinal and bulbar muscular atrophy. They have made use of previously described approach of expressing polyglutamine (polyQ) containing AR protein that contains either 24 or 100 polyQ repeats. These proteins were expressed under constitutive or inducible promoters. These mice were then used at different ages to study their muscle histopathology and cellular physiology. The findings reported validates many of the features previously reported in an independent mouse model with a single copy knock-in of AR with low, medium and high number of polyQ repeats. The findings here are mostly confirmatory of the findings from the previous study and the overall novelty of this work for the field is not too obvious. The authors would be better served clearly identifying this and stating it. A novel aspect of this study is the development of mice with inducible polyQ AR transgene. However, the authors have not fully explored the novelty of this model and have left this as the most poorly characterized mouse model in this study. At the very least the authors should run the same muscle histopathology studies and perform fiber type and fiber size measurements as in the other models.

One of the puzzling features of the findings here is the that despite mimicking the muscle mass and fiber type distribution seen in AR100Q, the AR24Q muscles do not exhibit the biochemical measure of atrophy and in fact shows increased PGC1a expression. These results contradict the claim that the biochemical indicators presented here are a consequence of the disease or indicate atrophic muscle loss. This also diminishes the impact of the mechanistic claims presented in this study. Further, the authors have not examined the status of CLCN1, an important feature of the polyQ AR knock-in mice.

Aside from the above major deficits, I find the study is well designed and the experiments are detailed and well described. However, there is a greater need for details such as sample sizes (number and age of animals used in each experiment) normality of the data and choice of statistical tests used. For example fiber analysis data do not report on how many mice were used. Furthermore, the authors make no effort to directly test for

Additional comments:

Introduction ends in a paragraph that appears to be some writing instruction or author comment, and should be deleted. Figures 6-8 do not seem to make marginally incremental points and represent little conceptual advance. Thus, these findings are best condensed and the concisely presented as a single figure. Figure 9 is anecdotal and lacks suitable quantification. This should be provided and there is a need for further experiment to validated the claimed interpretation of this finding. The discussion section is unduly long and should be condensed.

Author Response

Comments and Suggestions for Authors

This study describes generation of new mouse models for studying muscle pathology in SBMA - spinal and bulbar muscular atrophy. They have made use of previously described approach of expressing polyglutamine (polyQ) containing AR protein that contains either 24 or 100 polyQ repeats. These proteins were expressed under constitutive or inducible promoters. These mice were then used at different ages to study their muscle histopathology and cellular physiology. The findings reported validates many of the features previously reported in an independent mouse model with a single copy knock-in of AR with low, medium and high number of polyQ repeats. The findings here are mostly confirmatory of the findings from the previous study and the overall novelty of this work for the field is not too obvious. The authors would be better served clearly identifying this and stating it.

We agree with this reviewer that the novelty of our findings is not based on the generation of transgenic mice constitutively expressing polyQ-expanded AR. For this reason, we fully and deeply characterized the phenotype of the transgenic mice that we newly generated, and we showed that they recapitulate the main features observed in other mouse models of SBMA, i.e. premature death, motor dysfunction, muscle atrophy, denervation and aggregation. Nonetheless, the transgenic mouse model with constitutive expression of mutant AR has a very stable and consistent phenotype, and it has already been made available to other labs. Rather, a key point here was a comparison between the phenotype elicited by overexpression of normal and mutant AR. Overexpression of AR is per se toxic. However, our findings clearly show that, although both normal and mutant AR cause muscle atrophy, only mutant AR elicits a phenotype that resembles SBMA. To address the reviewer’s comment on novelty, we added a sentence to the last paragraph of the Introduction to explicitly point out that our deep characterization of the mouse phenotype recapitulates what has been shown in other mouse models, which per se is remarkably important as this represents a validation of the system. 

A novel aspect of this study is the development of mice with inducible polyQ AR transgene. However, the authors have not fully explored the novelty of this model and have left this as the most poorly characterized mouse model in this study. At the very least the authors should run the same muscle histopathology studies and perform fiber type and fiber size measurements as in the other models.

            We thank the reviewer for raising this point. To address it, we performed NADH and IF analysis of TA derived from iAR100Q mice treated with vehicle and dox (Figure 8G and Supplementary Figure 11A-B). Moreover, we analyzed the mean CSA of oxidative vs glycolytic fibers. We found that atrophy of glycolytic fibers exceeded that of oxidative fibers, thereby recapitulating an important aspect of SBMA muscle. However, two-week treatment of adult mice with dox was not sufficient to induce the metabolic alterations and fiber-type composition that are typical of SBMA muscle, thereby implying that expression of the disease protein during development or for a longer time period is necessary to elicit full disease manifestations. We changed our manuscript according to this new set of data.

One of the puzzling features of the findings here is the that despite mimicking the muscle mass and fiber type distribution seen in AR100Q, the AR24Q muscles do not exhibit the biochemical measure of atrophy and in fact shows increased PGC1a expression. These results contradict the claim that the biochemical indicators presented here are a consequence of the disease or indicate atrophic muscle loss. This also diminishes the impact of the mechanistic claims presented in this study.

Our results show that overexpression of both AR24Q and AR100Q results in a severe phenotype, which includes reduction of muscle mass and diminished body weight. However, our analysis of expression of markers of muscle atrophy, such as atrogenes, showed that atrogenes are upregulated specifically in AR100Q mice, and not in AR24Q. Moreover, mitochondrial pathology was evident in AR100Q mice, and not AR24Q mice. We propose that overexpression of AR is per se toxic, as previously shown by other groups (Monks et al., 2007), but expression of a polyQ-expanded AR triggers specific pathological processes in muscle, which are absent in AR24Q mice, and therefore specifically induced by the pathogenic expansion. For the sake of clarity, we better emphasized the aspects that distinguish the biochemical indicators of skeletal muscle atrophy (atrogenes and autophagy genes), as markers of SBMA muscle atrophy. We thank the reviewer for pointing this aspect out.

Further, the authors have not examined the status of CLCN1, an important feature of the polyQ AR knock-in mice.

            To address this point, we performed RT-PCR analysis of the transcript levels of Clcn1 in AR100Q mice and control mice. We found that Clcn1 transcript levels were downregulated in SBMA, but not control mice, at 8 weeks of age. These data are important, as in the first place they confirm that our transgenic SBMA mice reproduce another feature previously reported in knock-in SBMA mice, and in the second place they further support that altered muscle membrane excitability is a component of SBMA muscle pathology. These data are presented in Figure 4D of the revised manuscript. We thank this reviewer for this suggestion.

Aside from the above major deficits, I find the study is well designed and the experiments are detailed and well described. However, there is a greater need for details such as sample sizes (number and age of animals used in each experiment) normality of the data and choice of statistical tests used. For example fiber analysis data do not report on how many mice were used. Furthermore, the authors make no effort to directly test for

            We apologize for not including some statistical details of our experiments, and we thank the reviewer for the suggestion to incorporate them in the revised paper. Normality was tested using the Jarque–Bera test, a goodness-of-fit test of departure from normality, based on the sample skewness and kurtosis (Jarque et al., 1987). Using this test, normality was rejected only for the temporal curves [Hanging wire (HW), Rotarod for AR24Q and AR100Q, and body weight (BW)]. Given the lack of parametric tests able to handle missing values (premature death) and repeated measures, we recalculated the differences between AR100Q and WT mice using a two tailed Mann-Whitney test for each time point. All results were significant (p<0.05), the two groups began to differ already at 8 weeks of age for HW and at 7 weeks for BW. Furthermore, BW was smaller in AR100 Q relative to AR24 Q already at the age of 11 weeks. Impairment of rotarod performance was manifest already at 10 weeks. Lastly, for the 240 and 244-serine phosphorylated S6:total S6 differences in the quadriceps at 8 weeks normality was rejected and we used two tailed Mann-Whitney test to compare each genotype with wild type animals. We have detailed these changes 1) in the statistics section of the methods, 2) in the results, where we listed the tests used where a p-value was provided; 3) in the caption of each figure. We added the missing information to each Figure legend. In detail, we provided information related to the age and number of mice used in each panel. 

Additional comments:

Introduction ends in a paragraph that appears to be some writing instruction or author comment, and should be deleted.

We apologize for this oversight. We thank the reviewer for this comment.

Figures 6-8 do not seem to make marginally incremental points and represent little conceptual advance. Thus, these findings are best condensed and the concisely presented as a single figure.

            We combined Figures 6 and 8, and we moved Figure 7 to Supplementary information. We thank this reviewer, as we believe that this change improves the readability of our manuscript.

Figure 9 is anecdotal and lacks suitable quantification. This should be provided and there is a need for further experiment to validated the claimed interpretation of this finding.

            Quantification of the number of myonuclei with inclusion was added to the main text, page 12 of the revised manuscript.

The discussion section is unduly long and should be condensed. 

            We thank the reviewer for this comment. We shortened the Discussion, which allowed us to narrow down the focus of the manuscript on the main findings of this work.

Reviewer 2 Report

This extensive body of work by Marchioretti et al presents the generation of several new mouse models for the sex-specific and incurable disease, SBMA (aka Kennedy's Disease). There are currently a few models available for this disease, but this new set of mice will likely find an important place in the field, and potentially beyond, both to understand the biology of the disease as well as for the purposes of therapeutics development. Additionally, the Q24 mice might be of help to the cancer field, considering the huge importance of AR in prostate  malignancies.

To sum up a large body of work, the investigators present intriguing data, including a possible GOF from the over-expression (at very high levels) of the normal allele of the human AR protein, as well as polyQ-length dependent phenotypes in motorneurons and in muscles. The muscle phenotypes are particularly important since generally they have garnered less attention in SBMA, with MNs being thought of as of primary importance. Work from the Lieberman and Merry labs has placed muscles more front and center, but this work is also highly important in that direction. The importance of non-neural issues in SBMA might also help with similar studies in other polyQ diseases, which have generally ignored non-neuronal tissue even more so than the SBMA field.

From this work, the ARQ100 lines, both constitutive and conditional, will likely emerge as the more useful ones in the future. The ARQ24 findings are intriguing, concerning in some respects, and perhaps future studies will narrow down more why and how this specific line presents the way it does.

I find the work well conducted and reported. There are a few minor quibbles that the authors might want to address to help increase the reader's accessibility to the paper, especially for those who are not in the field. These points are listed below in the order they appeared to me; i.e. this order does not reflect significance:

Page 3: lines 106-112 do not belong here. They seem to be carryover from editorial directions. Figure 1 (and other similar presentations): it would help to have the legends for the lines in each panel. Figure S1: is the WT line missing? Or hiding behind another line? Kyphosis (page 3, line 239): photos, if available, would be nice.  Might help to clarify where the transgenes are inserted, especially considering the issues from Q24. For figures more generally: it is not always clear which sexes were tested. Perhaps this can be made clearer in legends as well as figures. Title of figure 2 does not seem to reflect well the findings from panel 2D. Are Q24 phenotypes due to where it is inserted? I.e. disruption of essential gene(s)? Any additional comments that might help in that direction? I hesitate to ask this considering the length of the Discussion, but i think it might be helpful in the future.  Discussion, helpful as it is at placing things into context, it might be trimmed a little. For example, the last two paragraphs could be combined.

Author Response

Reviewer 2

Comments and Suggestions for Authors

This extensive body of work by Marchioretti et al presents the generation of several new mouse models for the sex-specific and incurable disease, SBMA (aka Kennedy's Disease). There are currently a few models available for this disease, but this new set of mice will likely find an important place in the field, and potentially beyond, both to understand the biology of the disease as well as for the purposes of therapeutics development. Additionally, the Q24 mice might be of help to the cancer field, considering the huge importance of AR in prostate  malignancies. To sum up a large body of work, the investigators present intriguing data, including a possible GOF from the over-expression (at very high levels) of the normal allele of the human AR protein, as well as polyQ-length dependent phenotypes in motorneurons and in muscles. The muscle phenotypes are particularly important since generally they have garnered less attention in SBMA, with MNs being thought of as of primary importance. Work from the Lieberman and Merry labs has placed muscles more front and center, but this work is also highly important in that direction. The importance of non-neural issues in SBMA might also help with similar studies in other polyQ diseases, which have generally ignored non-neuronal tissue even more so than the SBMA field. From this work, the ARQ100 lines, both constitutive and conditional, will likely emerge as the more useful ones in the future. The ARQ24 findings are intriguing, concerning in some respects, and perhaps future studies will narrow down more why and how this specific line presents the way it does. I find the work well conducted and reported. There are a few minor quibbles that the authors might want to address to help increase the reader's accessibility to the paper, especially for those who are not in the field. These points are listed below in the order they appeared to me; i.e. this order does not reflect significance:

We thank this reviewer for these comments.

Page 3: lines 106-112 do not belong here. They seem to be carryover from editorial directions.

We apologize for this oversight. We edited the text accordingly.

Figure 1 (and other similar presentations): it would help to have the legends for the lines in each panel.

            We followed the reviewer’s suggestion and added the legend to each panel of Figure 1 and the other Figures of the manuscript.

Figure S1: is the WT line missing? Or hiding behind another line?

            The data relative to the WT mice are indeed hiding behind the data relative to the AR24Q mice.

Kyphosis (page 3, line 239): photos, if available, would be nice. 

We added pictures of WT and AR100Q mice to Supplementary Figure 1B.

Might help to clarify where the transgenes are inserted, especially considering the issues from Q24.

Please, see the response to the last comment.

For figures more generally: it is not always clear which sexes were tested. Perhaps this can be made clearer in legends as well as figures.

            We added the information relative to the sex of the animals used in each panel.

Title of figure 2 does not seem to reflect well the findings from panel 2D.

We modified the title of the Figure to point out that denervation is a late-onset event in this mouse model of SBMA.

Are Q24 phenotypes due to where it is inserted? I.e. disruption of essential gene(s)? Any additional comments that might help in that direction? I hesitate to ask this considering the length of the Discussion, but i think it might be helpful in the future. Discussion, helpful as it is at placing things into context, it might be trimmed a little. For example, the last two paragraphs could be combined.

            We deeply characterized the phenotype of AR24Q, and we found that the phenotype and pathology of this line resembles disease manifestations of mice overexpressing non-expanded AR, as shown by others (Monks et al., 2007). Although we cannot directly compare our mice to those mice reported previously, for several reasons, including the tg copy number inserted and the promoter used, we strongly believe that the phenotype observed in AR24Q mice results from overexpression of the tg, rather than from the site of insertion. Indeed, compared to the AR100Q mice, the AR24Q show signs of muscle pathology that are milder and therefore likely due to overexpression of an AR with a non-pathogenic repeat. However, we cannot rule out the possibility that the site of insertion contributes to the phenotype, and for this reason we will investigate it in the near future.  We reduced the length of the Discussion section, which allowed us to enhance the focus of this section on the main findings of this work, which was comparison of phenotype between overexpression of an AR with a short polyQ and with an elongated polyQ tract.

Monks, D.A., Johansen, J.A., Mo, K., Rao, P., Eagleson, B., Yu, Z., Lieberman, A.P., Breedlove, S.M., and Jordan, C.L. (2007). Overexpression of wild-type androgen receptor in muscle recapitulates polyglutamine disease. Proc Natl Acad Sci U S A 104, 18259-18264.

Jarque, Carlos M.; Bera, Anil K. (1987). "A test for normality of observations and regression residuals". International Statistical Review. 55 (2): 163–172.

Round 2

Reviewer 1 Report

The revised manuscript is significantly improved with the textual edits and inclusion of new data. This effort has not addressed the discord regarding the lack of bonafide biochemical indicators of atrophy in the AR24Q mice. Instead the authors have downplayed the statements in discussion and elsewhere that talks about this issue. Effort to better and more evenly present this aspect of the work and discuss the reasons for this disconnect will be of value to the readers. 

Author Response

We thank this reviewer for this further comment. We added a section to the Discussion to underly the relevance of this finding. We reported that lack of induction of these markers suggests a different pathogenetic mechanism underlying muscle atrophy in mice overexpressing normal AR. This warrants further investigation, which may help address the molecular link between AR GOF and muscle atrophy.